# Development of a Digital Lifestyle Modification Intervention for Use after Transient Ischaemic Attack or Minor Stroke: A Person-Based Approach

**DOI:** 10.3390/ijerph18094861

**Published:** 2021-05-02

**Authors:** Neil Heron, Seán R. O’Connor, Frank Kee, David R. Thompson, Neil Anderson, David Cutting, Margaret E. Cupples, Michael Donnelly

**Affiliations:** 1Centre for Public Health, Queen’s University Belfast, Belfast BT12 6BA, UK; s.oconnor@qub.ac.uk (S.R.O.); f.kee@qub.ac.uk (F.K.); m.cupples@qub.ac.uk (M.E.C.); michael.donnelly@qub.ac.uk (M.D.); 2School of Primary, Community and Social Care, Keele University, Staffordshire ST5 5BJ, UK; 3Royal Victoria Hospital, Belfast BT12 6BA, UK; 4School of Nursing and Midwifery, Queen’s University Belfast, Belfast BT9 7BL, UK; David.Thompson@qub.ac.uk; 5School of Electronics, Electrical Engineering and Computer Science, Queen’s University Belfast, Belfast BT9 5NB, UK; n.anderson@qub.ac.uk (N.A.); d.cutting@qub.ac.uk (D.C.)

**Keywords:** transient ischaemic attack, stroke, secondary prevention, risk reduction, behaviour change, physical activity

## Abstract

This paper describes the development of the ‘Brain-Fit’ app, a digital secondary prevention intervention designed for use in the early phase after transient ischaemic attack (TIA) or minor stroke. The aim of the study was to explore perceptions on usability and relevance of the app in order to maximise user engagement and sustainability. Using the theory- and evidence-informed person-based approach, initial planning included a scoping review of qualitative evidence to identify barriers and facilitators to use of digital interventions in people with cardiovascular conditions and two focus groups exploring experiences and support needs of people (*N* = 32) with a history of TIA or minor stroke. The scoping review and focus group data were analysed thematically and findings were used to produce guiding principles, a behavioural analysis and explanatory logic model for the intervention. Optimisation included an additional focus group (*N* = 12) and individual think-aloud interviews (*N* = 8) to explore perspectives on content and usability of a prototype app. Overall, thematic analysis highlighted uncertainty about increasing physical activity and concerns that fatigue might limit participation. Realistic goals and progressive increases in activity were seen as important to improving self-confidence and personal control. The app was seen as a useful and flexible resource. Participant feedback from the optimisation phase was used to make modifications to the app to maximise engagement, including simplification of the goal setting and daily data entry sections. Further studies are required to examine efficacy and cost-effectiveness of this novel digital intervention.

## 1. Introduction

Stroke is a leading cause of disability and mortality, for which a previous transient ischaemic attack (TIA) is a significant predictive factor [1,2]. An approximate 20% risk of a stroke event occurring within 90 days has been reported following a TIA, and early secondary preventative approaches are therefore recommended [3]. While these approaches include interventions targeted at reducing modifiable cardiovascular risk factors such as arterial hypertension and physical inactivity [4], individuals with higher cardiac risk frequently do not meet secondary prevention guidelines [5]. Early rehabilitative needs post TIA or minor stroke can also be complex [6,7], meaning it is uncertain how to best implement secondary preventative approaches in these populations.

In a recent systematic review examining aerobic and resistance exercise programs post TIA [8], of eight included trials, three reported change in time spent in moderate-to-vigorous physical activity, but only one showed significant increases in activity. While supervised aerobic exercise programmes have been associated with reductions in blood pressure following a TIA [9], such interventions are relatively time and resource intensive, and may not be feasible or cost effective in current healthcare system settings. Poor uptake and attendance, particularly among some groups, including older adults, can also limit their efficacy [10,11,12].

Mobile or mHealth-based apps may represent an accessible and cost-effective method for the delivery of lifestyle modification interventions post-TIA and stroke [13], particularly during the COVID-19 pandemic when social distancing is mandated [14]. These interventions can incorporate goal setting and pacing techniques [15], incentive and reward systems [16,17,18], and use of self-entered or automated data entry [19,20]. Adaptive elements such as tailored reminders or prompts [21], motivational messages [22] and responsive content [23] have also been explored. Despite this, there is comparatively little data on the core intervention components associated with engagement and usage needed for sustained behavioural change [24,25]. Many individuals discontinue use of mHealth interventions because of usability issues, or due to viewing them as irrelevant to their needs [26,27].

This paper describes the development of a digital lifestyle modification intervention for use in the early phases following a TIA or minor stroke (the ‘Brain-Fit’ app). This process was based on the person-based approach [28,29] and was informed by an analysis of the theoretical underpinning of the intervention and its likely mechanisms of effect. This app was modelled based on the ‘The Healthy Brain Rehabilitation Manual’ [30], a paper-based tool previously developed by the current authors for cardiovascular risk reduction following a TIA or minor stroke. In a randomised pilot feasibility trial, the tool was shown to be feasible, with potential for improvements in blood pressure and physical activity but qualitative data suggested that a digital version might improve accessibility and longer-term use [31]. We therefore aimed to explore perceptions on usability and relevance of the ‘Brain-Fit’ app in order to maximise user engagement and sustainability in TIA or minor stroke populations.

## 2. Materials and Methods

### 2.1. Study Design

Based on the person-based approach [28,29], a two-phase, iterative development process involved an initial ‘planning’ phase, and an ‘optimisation’ phase was used. ‘Planning’ refers to the triangulation of behavioural theories, qualitative review evidence and primary qualitative data to generate ‘guiding principles’ for the intervention, and logic modelling to describe key components and functions. ‘Optimisation’ refers to the use of qualitative data exploring participants’ views on a prototype intervention, including its acceptability and usability.

Phase I of the development process (intervention planning) included:A scoping review of qualitative evidence to identify perceived barriers and facilitators to use of digital health interventions in individuals with cardiovascular disease (CVD).Primary qualitative evidence from two focus group meetings to explore end users’ perceived needs and experiences after a TIA or stroke.Development of intervention guiding principles.Behavioural analysis and logic modelling.

Following development of a prototype version of the intervention, phase II (intervention optimisation) included:Primary qualitative evidence from an additional focus group to explore end user perspectives on the content and structure of the prototype, ‘Brain-Fit’ app.Qualitative think-aloud interviews to explore end users’ in-depth views on the prototype and its content, functionality and usability.

Ethical approval for the study was granted by the South West Cornwall and Plymouth Research Ethics committee (Ref: 19/SW/0213).

### 2.2. Phase I: Intervention Planning

#### 2.2.1. Scoping Review

The scoping review aimed to identify potential barriers and facilitators to user engagement and to successful implementation of digital interventions for people with CVD. A scoping review methodology was selected to explore and summarise a broad range of evidence and allow findings to be integrated into initial intervention planning. The review process followed an *a priori* protocol and was carried out using the Preferred Reporting Items for Systematic reviews and Meta-Analyses extension for Scoping Reviews (PRISMA-ScR) checklist [32], and following guidelines of the Joanna Briggs Institute Scoping Review Methodology Group [33]. Searches of four databases including Medline (via Ovid) were carried out by two authors (N.H., S.O.C.) from inception to 29 January 2020 using Medical subject headings (MeSH) (see Appendix A). Results were screened for studies reporting on experience of using digital interventions for lifestyle change in participants with established CVD. Following data extraction, findings on experiences of use, and barriers and facilitators were tabulated to inform the intervention ‘guiding principles’, behavioural analysis and logic model. Coding was used to identify key concepts and carry out a descriptive thematic analysis, based on an inductive approach [34].

#### 2.2.2. Primary Qualitative Evidence

Two focus group meetings were held in January 2020 involving members of patient support groups linked to a charity (Northern Ireland Chest Heart and Stroke (NICHS)). Forty-four individuals were invited to attend. Focus group methods are summarised in Table 1. Discussions explored views on the goals and needs of people following a TIA or minor stroke, their perspectives on exercise and physical activity and other lifestyle changes during recovery, and the specific role of digital interventions to support lifestyle change. For the purposes of this paper, only findings related directly to intervention development will be reported.

#### 2.2.3. Development of Intervention Guiding Principles

The person-based approach [28,29] emphasises the maximisation of user engagement and acceptability to increase intervention effectiveness and ease of implementation. The development of brief guiding principles is an important part of the overall approach. Guiding principles include design objectives or practice issues, and intervention components needed to address the objectives. Initial guiding principles were developed iteratively as evidence emerged from the scoping review, focus groups, analysis of the behaviour change componenets of the app and the perspectives of the study team.

#### 2.2.4. Behavioural Analysis and Logic Modelling

Key behaviours targeted by the ‘Brain Fit’ app, as well as the potential barriers and facilitators to use, were identified and coded using version 1 of the Behaviour Change Technique taxonomy (BCTTv1) [35], and mapped using the Behaviour Change Wheel (BCW) [36] and Theoretical Domains Framework (TDF) [37]. The BCTTv1 includes 93 techniques used to specify active components of behavioural interventions. The BCW is based on three central factors (capability, opportunity, and motivation) termed as the ‘COM-B system’ [38]. Domains of the TDF include knowledge and skills, beliefs, motivational factors and decision-making influences, and have been applied across healthcare settings to assist with design and evaluation, and examine the theoretical underpinning and behavioural influences of health interventions [37,39,40]. Using findings from this analysis, a logic model was developed to outline the proposed mechanisms of action of intervention components and anticipated outcomes. The preliminary logic model was reviewed iteratively to produce a final explanatory logic model.

#### 2.2.5. Prototype Intervention Building

Following phase I (intervention planning), a prototype ‘Brain-Fit’ app was built (version 1.0.17) and released as an early access version on the Google Play Store. This version underwent ongoing, iterative review by members of the project team to further develop its content according to phase I findings. The core content of the app is summarised in Table 2 and reflects that of the manual on which it is based [30,31].

### 2.3. Phase II. Intervention Optimisation

#### 2.3.1. Primary Qualitative Evidence from Additional Focus Group Meeting

An additional focus group was held in April 2020 as part of the ‘optimisation’ phase; and we used the same methods that were employed in the ‘planning’ phase (See Table 1). Individuals who were asked to attend a focus group meeting during phase I were invited again. Participants were presented with the content of the ‘Brain-Fit’ app on a large screen and mobile phones with the app installed were available. Discussions were focused on: presentation of the content and the information provided; use of goal setting/daily monitoring of physical activity using a pedometer; use of images and videos; use and content of app notifications and messages. Discussions also included aspects which were likely to be most successful in ensuring that it had good usability and factors that could promote long-term use. All potential and suggested changes were tabulated and subsequently, decisions were reached by study authors on whether to accept each change and include it in the revised version of the app (version 1.1.0). These decisions were made using the MoSCow method, a commonly used approach for prioritization of software requirements [41] and which assigns each potential change as (i) “Must Have” (Mo), (ii) “Should Have” (S), (iii) “Could Have” (Co), and (iv) “Won’t Have This Time” (W).

#### 2.3.2. Qualitative Think-Aloud Interviews

The final step in the optimisation phase, involved inviting all participants who had attended any of the three focus group meetings to take part in one-to-one think aloud interviews to explore in-depth views on the app, its content, functionality and usability. All interviews were conducted by the same researcher (S.O.C.) and took place in a community centre meeting room during March and April 2020. Participants were asked to spend approximately ten minutes using the app on a mobile phone, to become familiar with the app content and function: their use of the app was not monitored or recorded. Participants were then asked to complete a series of specific tasks, directed by the researcher. These included locating specific sections of the app, viewing videos, reading sections of text, and using interactive features such as the goal setting function and diary. Participants were asked to talk through the process of what they were doing and thinking during the tasks. The researcher used prompts when necessary. Approximately 20 min was set aside to discuss the app at the end of the tasks. Interviews lasted around 60 min in total and were audio recorded, with consent, transcribed verbatim and analysed initially by charting usability issues or barriers encountered. A thematic analysis approach was then used to explore views on the app and how it could be improved.

## 3. Results

### 3.1. Scoping Review

Of 2577 potential studies identified from database searches, 29 were included (see Appendix A). Three overarching themes relating to facilitators and barriers to user engagement and implementation of digital interventions emerged from the qualitative synthesis: (1) information and support needs; (2) motivational factors and health beliefs; and (3) technical and practical issues. The provision of easily accessible support and links to additional resources, including community-based programmes were key facilitators. Lack of accurate, condition specific knowledge was a frequently cited barrier. Motivation based on self-management approaches, goal setting, rewards, relatable information (including people in videos and images), and examples of others’ ‘shared’ experiences, were seen as being important to promoting routine engagement. Common barriers included increased anxiety, often linked to fear of recurrent cardiovascular events. Practical, simple, clear design and appropriate instructions were important, but unfamiliarity with digital technologies and lack of access to suitable devices were seen as barriers to uptake.

### 3.2. Primary Qualitative Evidence from Focus Group 1 and 2

The two initial focus group meetings were attended by 32 individuals (*N* = 18; focus group 2: *N* = 14) (see Figure 1 and Table 3). Three key themes emerged during analysis: (1) the role of lifestyle changes; (2) factors limiting participation; and (3) support needs during recovery. Lifestyle changes in the early stages of recovery focused on progressive increases in physical activity and were seen as an opportunity to increase confidence. Participants were also interested in the role of home-based exercises, such as walking on the spot, or stepping exercises, as these could be done easily at home without equipment, and were seen as a good ‘first step’ after a TIA or other cardiovascular event. Participants recognised the benefits of adopting a healthy diet but identified issues with knowing what dietary changes to make, and difficulties with maintaining any changes. Factors limiting participation, in terms of physical activity and exercise, included fatigue, persistent pain and the presence of co-morbidities. Some discussed the ‘invisible’ nature of having had a TIA or stroke, especially in later stages of recovery, and felt that this could lead to anxiety or embarrassment about not being able to be as active as before. Participants viewed digital interventions, in mobile, tablet or web-based formats as being useful and in particular, as ensuring provision of support and advice early after a TIA, especially when access to other sources of support was not available. Regarding support needs during recovery, participants discussed the different role of family members or partners. Some described feeling digital interventions should be shared with family members, to highlight how activity is ‘positive and low risk’, and to show information to help include them in physical activities. Social support from others was also seen as important and participants frequently suggested that information on “what to expect”, delivered via other patient stories would be useful. The findings from these focus groups concur with those of the scoping review. However, focus group participants did not indicate that unfamiliarity with digital technology or smartphones was a barrier to using the intervention with many reporting that they frequently used smartphones and apps, including some for monitoring physical activity.

### 3.3. Intervention Guiding Principles

The scoping review and initial focus group findings were used to develop preliminary guiding principles, presented in Table 4. These consist of four design objectives and intervention features included to address each objective. These principles were modified, based on emerging data from the iterative development and optimisation phases.

### 3.4. Behavioural Analysis and Logic Modelling

The behavioural analysis is shown in Appendix A. The intervention aimed to ensure effective engagement and support behavioural change with the main target behaviours being increased physical activity, uptake of a healthy diet, and monitoring of blood pressure. The content of the prototype intervention, coded using the BCT taxonomy, included 42 different BCTs from 14 sub-groups. All sources of behaviour within the BCW model were included in the intervention (reflective and automatic motivation, physical and psychological capability, physical and social opportunity, as well as five intervention functions (modelling, training, enablement, education, and persuasion). Eight behavioural domains on the TDF were targeted. These included: social influences, environmental context and resources, social role and identity, belief about capabilities, goals, emotion, knowledge, behavioural regulation, and physical skills. Figure 2 illustrates the intervention logic model. This included: identification of problems or gaps in care; targets for change (for lifestyle behaviours and mental health); intervention content; and proposed mechanisms of action and anticipated outcomes. This analysis indicated that to address target behaviours, the intervention must improve patient motivation and self-confidence. Potential effectiveness could be mediated via greater behavioural control and self-efficacy, facilitating more effective responses to barriers or setbacks when making lifestyle changes, and improving the capacity to apply behaviour change maintenance strategies, such as action planning.

### 3.5. Phase II. Intervention Optimisation

#### Primary Qualitative Evidence from Focus Group 3 and Think-Aloud Interviews

The third focus group meeting was conducted and attended by 12 participants. Ten participants, including six who took part in focus group 3, also agreed to take part in a think-aloud interview. Two individuals subsequently could not attend at the arranged time and therefore eight interviews were completed. Participants viewed the same prototype version of the app, without modifications being made between interviews (See Figure 3 for selected screen shots). Participants in later interviews were asked about their views on potential changes that had been suggested in earlier interviews. Participants had a broadly positive response to the prototype ‘Brain-Fit’ app (version 1.0.17). Due to the similarity in the content of the group discussions and think-aloud interviews during the optimisation phase, findings are summarised collectively. Three themes emerged from the overall synthesis of the findings: (1) using the information provided, (2) improving ease of use, and (3) providing ongoing support. Information was viewed as relevant and clear. The content about fatigue was valued by participants and included an acknowledgment of its potential impact on people. In terms of ease of use, participants were satisfied with the navigation and the look and design of the app; including the colours used, text and text size, and the use of chapters or sections.

Some expressed a view that text at the start of the app could be daunting to new users, as it appeared that lots of reading would be required. It was suggested that written content should be reduced, particularly at the beginning of sections to allow for a brief overview of each section to be given. These changes were made to ensure that the written content of the app was accessible and that it was understood easily by a broad range of users. The readability of the main sections of the app corresponded to a Flesch Kincaid Reading Ease score of 88, or a reading age of 11 or 12 years (https://www.webfx.com/tools/read-able/ (accessed on 23 April 2021).

Negative aspects influencing ease of use, and potential improvements, were identified during optimisation. This included simplification of the goal setting and manual data entry components, which were seen as too complex. Regarding ongoing support, participants liked the idea of using the app in different ways, both as a brief guide (without the depth of detail provided in some sections) or as a resource to review and re-read over time. Some participants indicated that re-reading the content could be an important motivational factor. The notes and comments section within the app was also seen as being a useful way to record thoughts and ideas to discuss with health professionals and this was highlighted as being important, especially when memory or cognition was affected post-TIA or stroke. This feedback resulted in a number of modifications being made to the revised version of the app. These changes are summarised in Appendix A.

## 4. Discussion

This paper outlines the systematic process of using a theory-, evidence-, and person-based approach to develop and optimise the ‘Brain-Fit’ app, a lifestyle modification intervention for use after a TIA or minor stroke. This study shows that the app-based behaviour change intervention has a good level of acceptability and could help to address gaps in support early after a TIA or minor stroke.

Some participants viewed fatigue as a significant barrier limiting their participation in physical activity. This aligns with previous evidence showing that fatigue can have a substantial impact on health-related quality of life, even after adjustment for other factors [31,42,43,44,45]. Another important influencing factor was anxiety or fear related to risk of recurrent TIA, or other cardiovascular events. This anxiety may result in individuals being unsure about, or not being motivated to make lifestyle changes such as increasing physical activity. ‘Reassurance’ provided by using the app in the early phases after a TIA was viewed as a potentially important mechanism to support behavioural changes.

The participants in this study identified the need for changes in physical activity behaviour to be made gradually. Realistic goal setting to facilitate progressive increases in physical activity may be central to restoring ‘confidence’ and providing a sense of personal control for individuals. Some studies, including a number of those in the scoping review within this study [46,47,48,49], have reported lack of familiarity with digital technology as a common barrier its use. However, our participants were generally comfortable with use of technologies, including activity monitors or trackers and apps used on smartphones or other devices. This may have been due to the relatively young age of participants in comparison to previous studies or that familiarity with technology has increased in the time since some studies in the review were published.

While there is currently limited evidence for the effectiveness of digital lifestyle modification interventions in TIA and minor stroke populations, previous reviews evaluating mobile apps targeted at promoting physical activity, indicate that behavioural techniques including goal setting, performance monitoring and feedback are common features [50]. Interventions, which include pedometers, have been shown to be effective in improving activity levels in individuals with increased cardiovascular risk [51], and were shown in this study to be acceptable for self-monitoring daily steps. Participants suggested that step data entry, using the goal-setting diary, and potentially discussing this with a health professional, was a particularly positive feature. This supports evidence indicating that participants value pedometer feedback on walking activity during cardiac rehabilitation and that using step goals can increase motivation to be physically active, and increase connection or relatedness to others. [52]

Practical advice on how to make lifestyle changes was seen as a potential means to improve patients’ motivation and self-confidence; suggesting that potential efficacy of the app, used with a pedometer, may be based on changes to behavioural control and self-efficacy, or a perceived ability to carry out a given behaviour [53]. Increased self-efficacy could result in more effective responses to barriers or setbacks when making lifestyle changes, with individuals more able to apply behaviour change maintenance strategies, such as action planning [54]. Goal setting and recording physical activity targets were also viewed as vital to support self-evaluation of progress based on measurable data.

The initial content of the ‘Brain-Fit’ app was modelled on a previous paper-based intervention [30,31], which had been developed using the MRC framework [55,56]. Use of the person-based approach [28,29] allowed for integration of current evidence and the perspectives of end users with this previous intervention in a systematic and iterative manner. Central to this process was the exploration and mapping of potential barriers and facilitators to effective user engagement and successful intervention implementation. This was informed by underpinning theories and theoretical frameworks, namely the Behaviour Change Technique Taxonomy [36], Behaviour Change Wheel [37] and the Theoretical Domains Framework [38] and user perspectives on digital interventions. Integration of extensive qualitative evidence provided for a detailed analysis of patient needs and current gaps in support and led to the inclusion of additional behaviour change components to those in the previous paper-based intervention [30]. These included ‘information from a credible source’, highlighting ‘Discrepancy between current behaviour and goals’ and additional use of ‘prompts or cues’ [36].

### Strengths and Limitations

A key strength of this study is that it followed a structured, iterative development and optimisation process. This study, based on the person-based approach [28] may provide information and a useful model for others developing digital interventions for this population. The researcher conducting the qualitative interviews during the study was independent of the initial development of the intervention, minimising potential bias. A potential weakness of the study is that the majority of participants were under the age of 70. This may have influenced responses on acceptability of mHealth interventions, and could have led to over-estimation of the familiarity of the target population with digital technology. The focus group meetings included between 12 and 18 participants and were therefore larger than is typically viewed as optimum [57]. There was also overlap with some participants involved in more than one stage during development. However, this may have been beneficial in that some who participated in the think-aloud interviews had familiarity with the app and were therefore able to provide additional insights and views on its function and usability.

## 5. Conclusions

Access to well-developed, evidence-informed digital interventions in the early stages after a TIA may represent an effective secondary preventative and rehabilitative approach to reduce cardiovascular risk. It is critical that apps provide accurate clinical information, and include features to ensure they have good usability, are engaging, and include effective behavioural components. Following these development and optimisation processes, the findings of a randomised controlled feasibility trial will be used to make any additional modifications needed to the ‘Brain-Fit’ app, prior to conducting a fully powered randomised controlled trial to examine intervention efficacy and cost effectiveness.

## Figures and Tables

**Figure 1 ijerph-18-04861-f001:**
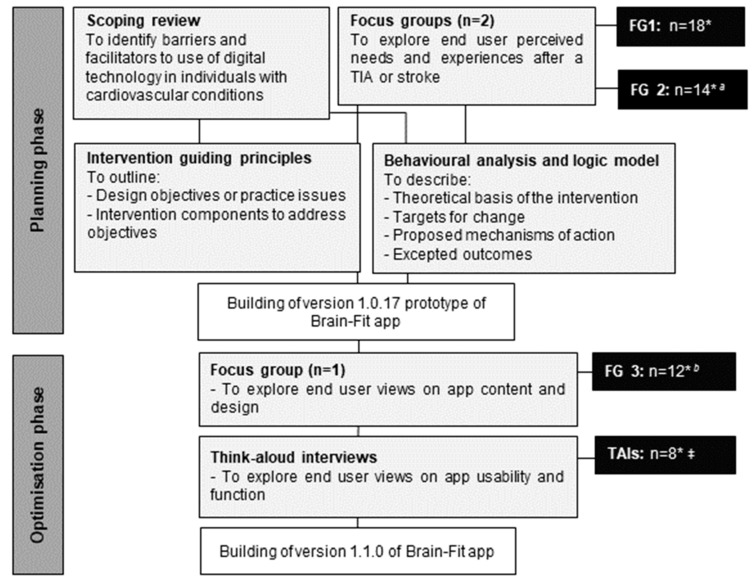
Stages in the process of developing the ‘Brain-Fit’ app. FG: focus group; TAIs: think-aloud interviews; * 32 participants took part in the study with some attending more than one focus group. Figures therefore reflect numbers attending at each step in the development process; ^a^ Of 14 participants at FG2, 5 had also attended FG1; ^b^ Of 12 participants at FG3, 7 had also attended FG1 or FG2; ^‡^ Participants in the think-aloud interviews had all taken part in one or more of the focus groups.

**Figure 2 ijerph-18-04861-f002:**
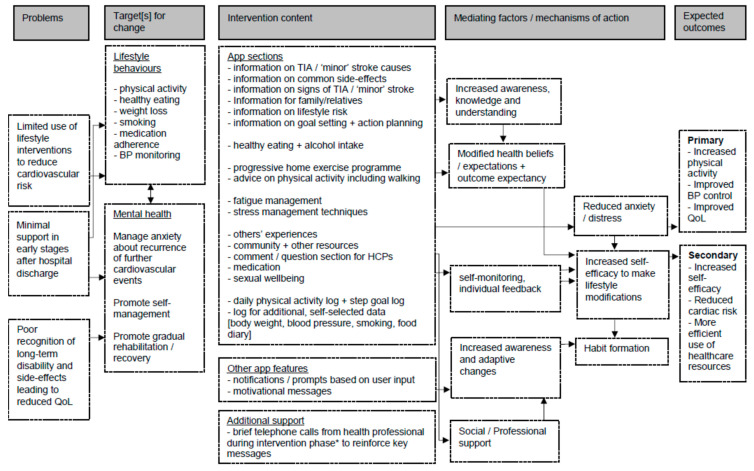
Logic model for mobile app to support patients after transient ischaemic attack or minor stroke (‘Brain-Fit’). QoL = quality of life; BP: blood pressure; HCP: Healthcare Proffessionals. * Included as part of intervention delivery during planned pilot studies.

**Figure 3 ijerph-18-04861-f003:**
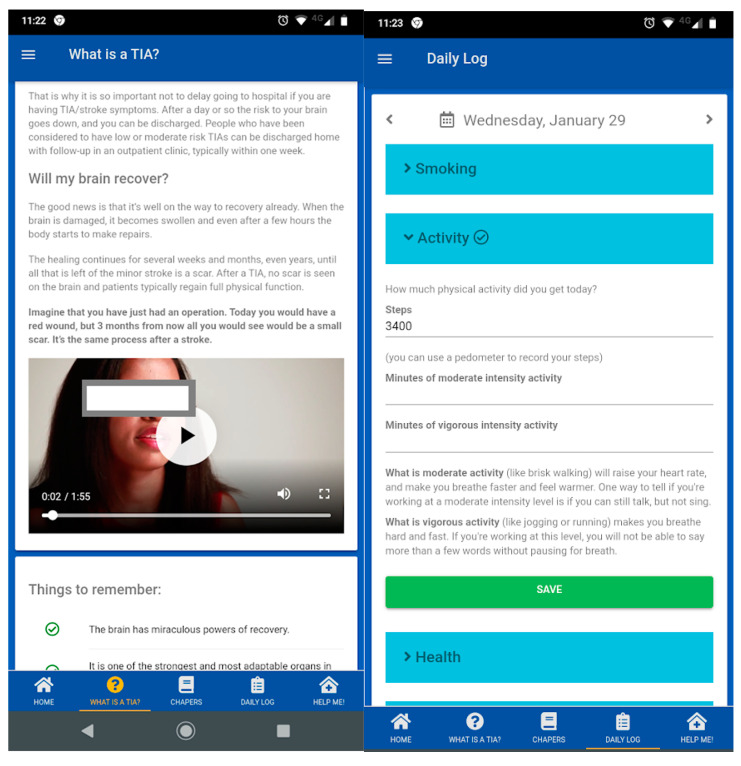
Selected screenshots showing content of the Brain-Fit app.

**Table 1 ijerph-18-04861-t001:** Summary of focus group methods used during the development and optimisation phases of the study.

Focus Group Methods
Individuals from patient support groups were invited in person or via telephone calls by a researcher (S.O.C) to attend a focus group meeting at a community centre. All participants had experienced a Transient Ischaemic Attack or stroke at least one month but no more than 2 years prior to recruitment. Focus group discussions followed a pre-determined schedule and lasted approximately two hours.
Discussions were audio-recorded and transcribed verbatim, with participants’ written, informed consent.
Transcribed data underwent a five-stage process during analysis. This included, data familiarisation, coding, generation of initial themes, review and definition of themes, and an analytical narrative synthesis to contextualise findings.
Transcripts were cross-coded by two authors (S.O.C. and N.H.) who met to discuss and resolve any disagreements.
Data were analysed using a reflexive, inductive thematic approach [34] with all decisions being discussed and confirmed via consensus between authors.

**Table 2 ijerph-18-04861-t002:** Core components of the ‘Brain-Fit’ app.

Intervention Components
Chapters Introduction General information on TIA and minor stroke - Goal setting - Action planning - Examples of action planning - Information on recognising signs of a TIA or stroke Be more active, more often - Benefits of physical activity - What is exercise and physical activity? - Recommended amount of exercise and physical activity? - Warming up before exercise and physical activity - Exercises to try at home - Exercise and physical activity diary Having a healthy diet - Recommended diet - Setting targets People’s experiences after a TIA or minor stroke Managing stress - What is stress? - Hints and tips for managing stress Managing fatigue - Hints for managing fatigue - Other treatments for fatigue Stopping smoking - Benefits of stopping smoking - Preparing to quit - Hints and tips to help stop smoking Medication Sex after a TIA or minor stroke Community resources General information on employment, driving and relevant sources of support.
Daily log - Step goal reminder and entry - Step data entry
My notes and reminders - Manual text entry
Other app features [automated] - notifications/prompts - motivational messages

**Table 3 ijerph-18-04861-t003:** Demographic details of participants included in the study.

	Total Participants (N= 32)
Age category	
18–49 years 50–69 years 70+ years	3 (9.4%) 23 (71.8%) 6 (18.7%)
Gender	
Female Male	21 (65.6%) 11 (34.4%)
Ethnicity	
White Black/African/Caribbean/Black British Other	30 (93.7%) 1 (3.1%) 1 (3.1%)
Previous use of mobile technology (smartphone/tablet)	
Yes No	27 (84.4%) 5 (15.6%)

**Table 4 ijerph-18-04861-t004:** Intervention guiding principles showing design objectives and key intervention features.

	Design Objectives	Key Intervention Features
1	Increase confidence and self-efficacy for making behavioural change and address barriers to lifestyle change Barriers addressed: 1, 2, 3, 4, 5, 6, 7, 8 *	- Promote changes in behaviour including gradual, progressive increase physical activity. - Provide a range of goal setting examples. - Include use of goal setting techniques and step data entry (from manual pedometer or a pedometer app). - Include patient stories detailing experiences of recovery after a TIA or minor stroke. - Include information on benefits of healthy behaviours with minimal reference to risk avoidance. - Include additional support from a healthcare professional (telephone calls) to reinforce key messages and provide further reassurance.
2	Ensure ease of use and good intervention acceptability Barriers addressed: 9, 10, 13 *	- Provide different levels of detail including short summaries of each section as well as longer, more detailed sections. - Include health professional telephone support as part of intervention.
3	Provide accessible, brief information and support that can be viewed easily on mobile devices (promoting frequent/daily use) Barriers addressed: 1, 8, 9, 11, 12 *	- Include simple, clear information and language. - Provide advice on integrating the app into a daily routine. - Use of visual aids, graphics and media.
4	Promote self-management and longer-term behavioural change Barriers addressed: 1, 2, 5, 11 *	- Include motivational messages and prompts. - Include a diary/notes section. - Promote choice through self-selection of additional target behaviours (e., diet, smoking, stress management, medication) - Include links to additional sources of support and localised resources (both digital and non-digital resources) - Include references to support/involvement of family/friends in recovery.

* Barriers identified during scoping review and focus groups. 1. Lack of information on cardiovascular conditions including causes/risk factors. 2. Concern that TIA is an ‘invisible’ condition and others including health professionals perceive effects as minor. 3. Fatigue. 4. Pain/mobility limitations. 5. Lack of confidence or ‘self-belief’. 6. Focus on other aspects of care and recovery. 7. Anxiety about recurrence. 8. Uncertainty how to adopt/ maintain a healthy diet. 9. Limited access to digital interventions. 10. Unfamiliarity with digital technologies/smartphones. 11. Family/partner role (perceived ‘protective’ role). 12. Poor weather or lack of places to be active. 13. Effect on memory or cognition.

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
