# Peer review of "Development of a Digital Lifestyle Modification Intervention for Use after Transient Ischaemic Attack or Minor Stroke: A Person-Based Approach"

_ijerph, 2021, doi:10.3390/ijerph18094861_

Round 1

Reviewer 1 Report

Thank you for the opportunity to review the manuscript by Heron et al on development of app, as a secondary prevention intervention post stroke or TIA, to address the gaps in support and enhance people participation for life style modifications. This is an extremely well written manuscript with good methodology and a very structured approach to development of this app. I congratulate the authors.

I think the paper was so well written and methodology was so solid that I did not have much critique. The major strength of this manuscript is the structured process of development and optimization of the app, modelled on "person based approach".   Maintaining engagement can be especially difficult when "digital behavior change interventions" are used without human support which leads to high levels of dropout and attrition. Central to this process is the exploration and mapping of potential barriers and  facilitators to effective user engagement and successful intervention implementation. This manuscript reflects on how interventions can be designed to fit the user and their specific needs and context.

My only comment for authors would be if any efforts were made to ensuring they are accessible to those with lower levels of education or income?

Reviewer 2 Report

This is a nice systematic report about the development of an app-based intervention to invoke behavioral change in acute TIA-survivors. The authors do a nice job in reporting background for concept development and qualitative research design. There are a few editorial modifications which can help the paper in terms of making it a stronger statement of work. Please see attached document with comments throughout. 

Reviewer 3 Report

The authors describe the development of a secondary prevention intervention in the form of a mobile app for patients having suffered from transient ischemic attack or minor stroke. The paper is very well written, and each step is described in details. The whole development process has the strength of focusing on end-users needs by including them all along. I congratulate the authors for this very nice and complete work.

Minor comments:

1. In the methods, under 2.3.2 "Qualitative think-aloud interviews", I find it quite confusing when you specify the number of patients (32) who had attended one of the focus group. I think this number should only appear in your results as at this point we don't know exactly how many patients participated in you study.

2. I think there might be an error (line 182) when you mention during April and March 2020. Is it really March? In this case it would be more logical to mention it prior to April.
